# Adversarial Example Quality Assessment: A Large-scale Dataset and Strong Baseline

## ABSTRACT

Adversarial examples (AEs), which are maliciously hand-crafted by adding perturbations to benign images, reveal the vulnerability of deep neural networks (DNNs) and have been used as a benchmark for evaluating model robustness. With great efforts have been devoted to generating AEs with stronger attack ability, the visual quality of AEs is generally neglected in previous studies. The lack of a good quality measure of AEs makes it very hard to compare the relative merits of attack techniques and is hindering technological advancement. How to evaluate the visual quality of AEs remains an understudied and unsolved problem. In this work, we make the first attempt to fill the gap by presenting an image quality assessment method specifically designed for AEs. Towards this goal, we first construct a new database, called AdvDB, developed on diverse adversarial examples with elaborated annotations. We also propose a detection-based structural similarity index (AdvDSS) for adversarial example perceptual quality assessment. Specifically, the visual saliency for capturing the near-threshold adversarial distortions is first detected via human visual system (HVS) techniques and then the structural similarity is extracted to predict the quality score. Moreover, we further propose AEQA for overall adversarial example quality assessment by integrating the perceptual quality and attack intensity of AEs. Extensive experiments validate that the proposed AdvDSS achieves state-of-the-art performance which is more consistent with human opinions.

## CCS CONCEPTS

• **Computing methodologies** → **Appearance and texture representations**; • **Theory of computation** → **Adversarial learning**.

## KEYWORDS

Adversarial attack; Adversarial examples; Image quality assessment; Human visual system

## 1 INTRODUCTION

The vulnerability of deep neural networks (DNNs) to adversarial examples, *i.e.*, hand-crafted images that are similar-looking to clean images but can induce dramatic changes in DNNs, has raised an increasing threat to safe-critical applications, such as face verification [27], object tracking [35], and self-driving [6]. This intriguing

discovery has attracted growing attention and spawned considerable research for adversarial attacks. In recent years, numerous adversarial attack methods have been proposed, including gradient-based [5, 10, 13, 14, 18, 23, 30], optimization-based [4, 29], and learning-based attacks [1, 24, 33].

As a general rule, adversarial attacks generate adversarial examples to fool the target model by pursuing two goals: (1) *Invisibility*: Perceptually, the adversarial examples should be similar-looking to the benign images; (2) *Lethality*: In the target models, the distribution of generated adversarial examples should differ from that of benign images and trigger wrong prediction behaviors of DNN models. However, while numerous efforts are devoted to developing adversarial examples with stronger attack ability, the perceptual quality of adversarial examples is generally neglected. Almost all the existing methods adopted $L_p$-norms (e.g., $L_1$, $L_2$, and $L_\infty$) as the constraint to keep the generated adversarial examples similar-looking to the original image; however, the norm-based constraint does not correlate with human perceptual quality since it computes the pixel-wise error and does not include the properties of human visual system (HVS). We present a diagnosis report of the perceptual quality of AEs in Table 1, where we can see that the $L_p$-norm cannot well-assist the perceptual quality of AEs, and different attack strategies can produce AEs with different perceptual qualities. The lack of good quality measurement of adversarial examples makes it very hard to compare the relative merits of adversarial attack techniques.

On the other hand, image quality assessment (IQA) has been an active research topic in the last decades [7, 15, 16, 20, 25, 31, 39, 41]. An intuitive way is to directly take advantage of existing IQA metrics; however, most of the existing IQA metrics are designed to capture certain distortions, such as blur and color fading for de-fogging [7, 16], edge sharpness for super-resolution [41], content compensation for tone-mapping [20]. Since adversarial perturbations have different properties from those applications, using these IQA metrics can lead to unsatisfied results (see the fifth row of Table 1). Developing a new reliable metric specifically designed for Adversarial examples remains an unsolved and challenging problem.

In this paper, we make the first attempt to fill a gap in the literature by presenting an image quality assessment metric specifically designed for evaluating the perceptual quality of adversarial examples. To achieve this goal, we face two challenges: First, a suitable adversarial example quality assessment metric should reflect the degradation of the image that is consistent with human subjective evaluation, e.g., the mean opinion score (MOS). However, there is no such off-the-peg dataset for evaluation; Second, the perceptual properties of adversarial examples are unknown. Since the existing attacks have already used $L_p$-norm constraint, the adversarial perturbations are near-threshold (i.e., just-visible to human eyes), which is hard to capture and quantified.

**AEQA diagnosis report**

| Adversarial Examples (AEs) |  |  |  |  |  |
|---|---|---|---|---|---|
| Attack algorithm | FGSM | PGD | BIM | MIM | PGD |
| $L_2$-norm constraint | 0.03 | 0.03 | 0.03 | 0.03 | 0.4 |
| Typical IQA metrics (PSNR, NIQE) | (30.07, 49.26) | (35.13, 11.43) | (**35.51**, **7.58**) | (31.23, 47.19) | (13.45, 21.66) |
| AdvDSS | 0.9012 | **0.9741** | 0.9625 | 0.9236 | 0.3334 |
| **MOS** (subjective evaluation) | 2.9801 | **3.6742** | 3.3351 | 3.2220 | 2.0301 |

Table 1: A diagnosis report of adversarial examples produced by different adversarial attacks. From the table, we can have the following observations: 1) The $L_2$-norm constraint cannot well assist the *invisibility* of AEs, and different attacks with the same $L_2$-norm constraint still have different results; 2) The adversarial distortion is near-threshold that is just-visible to human eyes; 3) Typical IQA metric can hardly capture the distortion in AEs, e.g., the AE produced by PGD has a better visual quality than BIM but achieves a lower PSNR value.

To address these challenges, we first construct a new annotated adversarial example quality assessment database, named "AdvDB", focusing on adversarial examples with diverse perturbations, which is the first and largest database for assessing adversarial examples. Specifically, we collected 18, 346 adversarial examples produced by 5 dominant adversarial attack algorithms with different parameter settings, and each image is annotated with two attributes by 11 professional subjects. Based on the AdvDB, we conducted analysis and found that the structural information can reveal the perceptual quality of adversarial examples, thus we further proposed a visual Detection-based Structural Similarity index (AdvDSS) for assessing the perceptual quality of adversarial examples. To capture the near-threshold adversarial distortions from the perceptual perspective, AdvDSS employs a detection scheme that incorporates HVS technology to first compute the visual saliency error map and then conduct structural similarity comparison to predict the perceptual scores. Moreover, by integrating the perceptual quality and attack strength of adversarial examples, we take a further step and present an adversarial example quality assessment (AEQA) for comprehensively evaluate the quality of adversarial examples.

In a nutshell, our contributions can be summarized as follows:

- We build so far, the largest annotated adversarial example quality assessment database named "AdvDB". Specifically, we collected 18, 346 adversarial examples produced by 5 dominant adversarial attack algorithms based on 3 DNN backbones with different parameter settings, and each image is annotated with two attributes by 16 professional subjects.
- Based on the data analysis of AdvDB, We propose a visual Detection-based Structural Similarity index (AdvDSS) for assessing the perceptual quality of adversarial examples.
- By integrating the distribution divergence of adversarial examples, we take a further step and present a novel benchmark adversarial example quality assessment metric (AEQA)

for comprehensively evaluating the quality of adversarial examples.

- Extensive experiments illustrate that our detection-based approach achieves state-of-the-art performance and is also consistent with human visual systems. We also investigate various applications of AEQA in comparing and improving existing adversarial attacks.

## 2 RELATED WORK

### 2.1 Adversarial Attack Techniques

There have been extensive explorations for the generation of adversarial attacks in the literature, which can be categorized into gradient-based methods [10, 13, 14, 18, 23, 30], optimization-based methods [4, 29], and learning-based methods [1, 33]. The gradient-based methods originated from [14] where Goodfellow *et al.* proved that the existence of adversarial examples is the result of the linearity of DNN models and proposed Fast Gradient Sign Method (FGSM) to generate perturbations by the model gradient change. Following works enhance the attack ability by breaking the one-step gradient into iterative generation [18, 23, 30], local enhancement [13, 37], and data augmentation [10, 34]. The gradient-based attacks exhibit high attack success rates but require explicit knowledge about the target model (i.e., architecture and parameters). In contrast, optimization-based adversarial attacks generate perturbations via optimization algorithms. Szegedy *et al.* [29] proposed L-BFGS method to generate adversarial examples by simultaneously optimizing the misclassification and perceptual deviation. Carlini *et al.* [4] proposed a CW attack by constraining the number of clean examples changing, the overall degree of perturbation, and the maximum allowed perturbed per pixel by $L_0$, $L_1$, and $L_\infty$ distance, respectively. The learning-based methods utilize generative models to directly transform original images into adversarial examples.

Baluja *et al.* [1] proposed Adversarial Transformation Networks (ATNs) to transform an original image into an adversarial example via a generative model by simultaneously increasing the attack rate and constraining the similarity deviation. Xiao *et al.* [33] further provided AdvGAN to utilize a discriminator for better visual quality. These methods can achieve high attack success rates; however, the perceptual quality of the generated adversarial examples is neglected.

## 2.2 Image Quality Assessment

Image quality assessment (IQA) has been an active research topic in the last decades [2, 3, 15, 19, 22, 22, 25, 28, 39]. IQA can be divided into full-reference (FR)-IQA and non-reference (NR)-IQA, with FR-IQA has a reference image and the goal is to quantify the visual differences between reference and test images, and NR-IQA does not include reference images and aims to produce subjective human appreciation based on the quality of image content. Given the purpose of quantifying the adversarial distortions over a reference image in our work, we focus on FR-IQA here. Existing FR-IQA methods are either general-purpose methods [9, 11, 31, 39, 40] or distortion-specific methods [7, 16, 20, 41]. In the general-purpose methods, the assessment is conducted by extracting natural statistical characteristics of images, including traditional methods PSNR, SSIM [31], MAD [11], FSIM [39], and deep learning based LPIPS [40], PieAPP[25], and DISTS[9]. These methods can evaluate the general quality of images but can not explicitly quantify certain distortions. In contrast, the distortion-specific methods evaluate the image quality by using the features of known distortion type, such as FRFSIM [16] and FADE [7] for defogging, SRIF [41] for image super-resolution, and HDR-VDP-2 [20] for tone-mapping. Despite remarkable progress have been made in FR-IQA, adversarial distortion is seldom explored in the literature.

## 2.3 Adversarial Example Quality Assessment

The perceptual evaluation of adversarial examples remains an open problem in the field. To our best knowledge, there have not yet developed any metrics specifically designed for adversarial examples. There is a prior work [12] that conducts an analysis of adversarial example perceptual quality, where the authors built a small-sized database and investigated the existing IQA methods in evaluating adversarial examples. However, the database only contains 13 images which is limited for evaluation, and a metric specifically designed for adversarial examples is still lacking. In this paper, we make the first attempt and focus on distortion-specific FR-IQA for evaluating the perceptual quality of adversarial examples.

## 3 ADVDB: ADVERSARIAL EXAMPLE ASSESSMENT DATABASE

In this section, we introduce the details of the proposed database, including the database construction and analysis.

### 3.1 Database Construction

**Collection.** Considering our work focuses on the adversarial examples in the classification task, we choose samples from the classic databases for classification, namely CIFAR-10/100 and ImageNet databases. Specifically, we randomly choose $1,500$ samples from

**Table 2: Parameter settings in adversarial example generation during data collection.**

| Attack | $L_p$-norm | Parameter | Values |
|---|---|---|---|
| PGD [18] | $L_0, L_2, L_\infty$ | $\varepsilon$ | 0.003, 0.03, 0.1, 0.4, 1.4 |
| FGSM [14] | $L_0, L_2, L_\infty$ | $\varepsilon$ | 0.002, 0.03, 0.06, 0.14, 0.4 |
| BIM [30] | $L_0, L_2, L_\infty$ | $\varepsilon$ | 0.003, 0.03, 0.06, 0.15, 0.3 |
| MIM [10] | $L_0, L_2, L_\infty$ | $\varepsilon$ | 0.005, 0.03, 0.06, 0.19, 0.6 |
| Deepfool [23] | $L_0, L_2, L_\infty$ | overshoot | 0.25, 1.0, 3.5, 36, 500 |

these datasets, which covers $1,000$ classes of objects with a wide range of colors and textures under both indoor and outdoor scenes. To make the images more applicable for generating adversarial examples, we cropped the images into the size of $299 \times 299$ covering the main object in the image.

**Adversarial Example Generation.** Having the reference images, next we propose to generate adversarial examples by applying adversarial attacks on these clean images. Five prominent attacks are employed in this process, namely FGSM, BIM, Deepfool, PGD and MIM attacks. Note that each attack can be tuned into different attack strengths via a set of parameters. We fine-tune the parameters of each attack to generate diverse adversarial examples, from imperceptible levels to overt levels. As for the source model, we consider Inception v3 network pre-trained on ImageNet dataset to produce adversarial perturbations. We list the parameters used in each attack in Table 2. To ensure the generated AEs have attack ability, we filter out those AEs with correct prediction results. Consequently, we obtain $18,346$ adversarial examples that can succesfully attack the target model using five attacks with various parameter settings.

**Annotation.** We invited 11 professional researchers to perform subjective evaluation by comparing the adversarial examples and their corresponding reference images. The subjective software was conducted under the same environment with a 4K Ultra HD LED Monitor. In addition, the subjects are given instructions before the experiments to circumvent bias in this task. As shown in Figure 1, a customized interface is designed to render a sequence of images simultaneously. The left image is the original reference image, and images 1-5 are adversarial examples generated by different attack algorithm settings in random order. The subjects are asked to give a score from 1 to 5 for each adversarial example based on the perceptual possibility of this image being attacked. A higher score means that the image is perceptually abnormal, which denotes a higher possibility that the image is perturbed. Finally, the outlier individual score is removed and then the remaining scores are averaged as the MOS of each adversarial example.

### 3.2 Data Analysis

**Annotation Distribution.** The AdvDB consists of $18,346$ adversarial examples, covering 75 attack strategies, and quality scores from 1 to 5, as shown in Figure 2. It can be observed that the distribution of attack strategies is even, with a slight decrease when $\epsilon \leq 0.01$. This is because the adversarial examples cannot successfully fool

Anonymous Authors

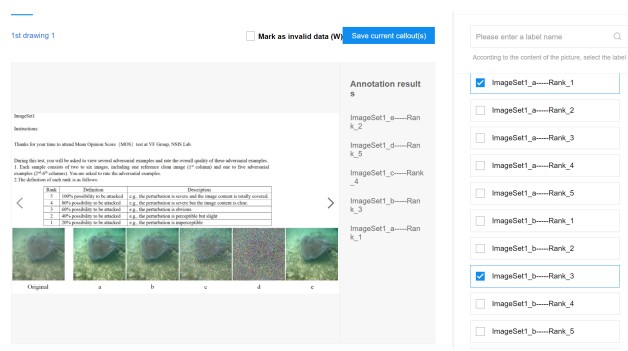

Figure 1: Screenshot of the subjective evaluation interface.

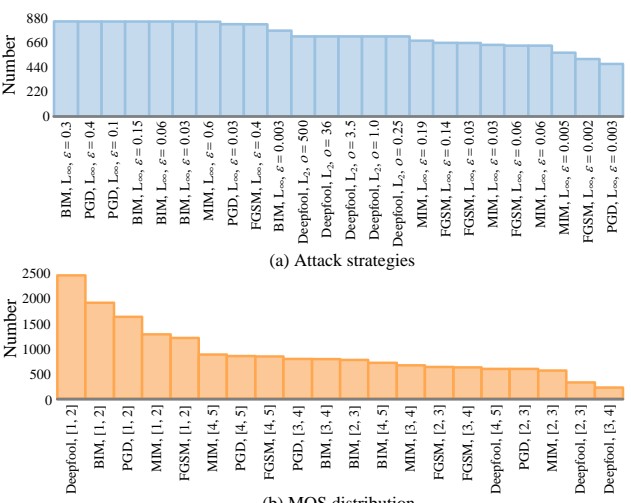

(a) Attack strategies

(b) MOS distribution

Figure 2: Statistics of AdvDB. (a) The adversarial attack strategy distribution. (b) MOS distribution of all the adversarial examples.

the target model when $\epsilon$ is too small so we filter out these examples during database construction. From the MOS distribution, we can see that the score $[1, 2]$ takes a high frequency in the whole database. It indicates that adversarial perturbations are always just-visible to human eyes since they have already been constraint by $L_p$-norm during generation.

**FR-IQA Preference.** We conduct a correlation analysis between the subjective results and existing popular FR-IQA metrics. Specifically, we adopt the most often used metrics in image quality assessment including PSNR, SSIM, NIQE, MAD, LPIPS, DISTS, and GMSD. Also, the three widely used distance norms $L_1$, $L_2$, and $L_\infty$ in adversarial attacks are also considered. We use Spearman's rank order correlation coefficient (SROCC), Pearson linear correlation coefficient (PLCC), Kendall rank order correlation coefficient (KROCC), and root mean squared error (RMSE) to evaluate the correlations. Note that higher SROCC, PLCC, KROCC values and lower RMSE value denote stronger correlations. The results are shown in Figure 3. We can have three observations here: First, the $L_p$ distance cannot well assess the perceptual quality of AEs as the $L_p$ distance generally achieve low correlations with MOS. Second, almost all the FR-IQA metrics have positive correlations with the obtained subjective annotations, where most SROCC and PLCC values are higher than 0.9. This indicates the proper design and conduction of our subjective annotations. Third, among all the FR-IQA metrics, SSIM has the strongest correlations with subjective evaluations for adversarial examples, indicating that the structural information can reveal the adversarial distortion in adversarial examples, which inspires us to utilize the structural similarity to perform perceptual assessment of AEs.

## 4 APPROACH

In this section, we introduce our approach, whose main idea is to construct a benchmark image quality assessment metric that is specifically designed for AEs following an overall quality metric for comprehensively evaluating AEs from both perceptual quality and attack intensity.

### 4.1 Perceptual Quality Assessment of Adversarial Examples

**Design.** Here our goal is to develop a perceptual quality assessment for adversarial examples. We first revisit the formulation of AEs which is described as:

$$x^{adv} = x + \delta, \text{ s.t. } x^{adv} \in \mathcal{B}_\epsilon(x), \tag{1}$$

where $x$, $\delta$, and $x^{adv}$ denotes clean image, adversarial perturbation, and AE, respectively. $\mathcal{B}_\epsilon(x)$ denotes the $\ell_p$-norm ball centered at $x$ with radius $\epsilon$, i.e., $\mathcal{B}_\epsilon(x) = \{x^{adv} : \|x^{adv} - x\|_p \leq \epsilon\}$. Existing advances in adversarial attacks try to find a proper $\delta$ to make the model loss maximized. Since the AEs are already constrained by $L_p$-norm, the distortions over benign images are near-threshold, which brings challenges in evaluation. Thus we consider the design of the metric from two aspects: (i) It should reflect the image degradation that is consistent with MOS thus HVS should be incorporated; (ii) It has been pointed out that human visual system tends to locate visible differences in near-threshold distorted images for giving the MOS while attempts to recognize image content when faced very low-quality images. Based on this prior, we employ a detection scheme where the visual saliency is first detected and then the differences are further quantified by conducting structural similarity evaluation. An overall overview of the whole process is shown in Figure 4.

#### 4.1.1 Visual Saliency Detection. We first conduct visual saliency detection. This phase is composed of two steps, gamma correction and contrast sensitivity function transformation.

**Gamma Correction.** Let $x$ denotes a clean image, and $x^{adv}$ denotes a corresponding adversarial example. We first transform the images from RGB space to YCbCr space for adapting the human perceptual system. According to the HVS theory [11], human eyes are more sensitive to darker regions in images and less sensitive to brighter

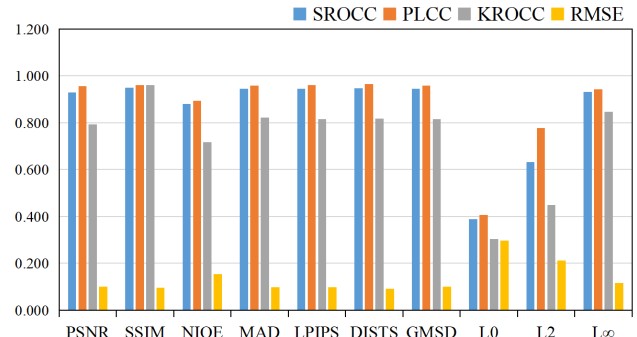

Figure 3: Correlations of MOS and existing FR-IQA methods on the adversarial examples in AdvDB.

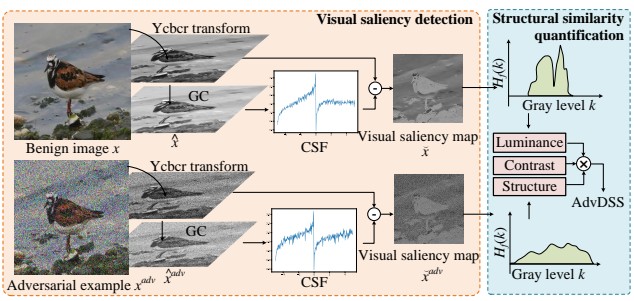

Figure 4: Overview of the AdvDSS procedure where the visual saliency is first detected and the structural similarity between the AE and benign image is quantified to produce the AdvDSS score.

regions. Thus, we employ gamma correction (GC) to preserve as many dark details as possible in the images, which is defined as:

$$\hat{x} = (\frac{x}{255})^{\gamma} * 255, \tag{2}$$

where $\gamma$ is empirically set to be $\frac{1}{2.2}$ as suggested in previous works. Note that the GC is applied to both $x$ and $x^{adv}$ for better quantification.

**Contrast Sensitivity Function Transformation.** Human eyes are believed to be sensitive to the spatial frequency of images. To account for variations in different spatial frequencies, we adopt the contrast sensitivity function (CSF) proposed by Mannos and Sakrison[19] with adjustments specified by Daly[8], which is calculated as

$$H(f, \theta) = \begin{cases} k(c + \lambda f_\theta) \exp[-(\lambda f_\theta)^{1.1}], \text{if } f \geq f_{peak}c/deg \\ 0.981 \qquad\qquad otherwise \end{cases}, \tag{3}$$

Where $f$ is the radial spatial frequency in cycles per degree of visual angle (i.e., $c/deg$), $\theta$ is the orientation and varies in $[-\pi, \pi]$, and $f_\theta = f/[0.15cos(4\theta) + 0.85]$ is an orientation-based modification of $f$ that along the diagonal orientations to account for decreased contrast sensitivity. Note that we adopted the adjusted CSF which has a lowpass by setting frequencies below $f_{peak}$ to 0.981. Following previous works [8], the $f$ and $\theta$ can be computed from discrete Fourier

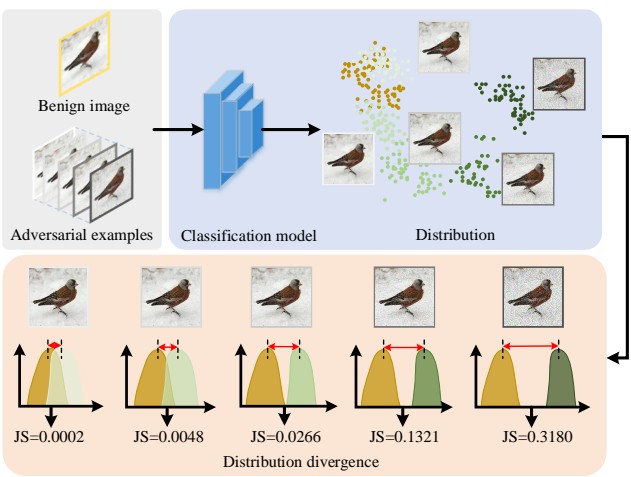

Figure 5: Illustration of the attack intensity quantification using JS divergence between clean images and adversarial examples.

transform (DFT) indices $u \in [-M/2, M/2]$ and $v \in [-N/2, N/2]$ via:

$$f = [(\frac{u}{M/2})^2 + (\frac{v}{N/2})^2]^{\frac{1}{2}} \frac{\rho \vartheta \tan(\frac{\pi}{180})}{2}c/deg, \tag{4}$$

$$\theta = \arctan(\frac{v}{u}), \tag{5}$$

where $\rho \vartheta \tan(\frac{\pi}{180})$ is the display visual resolution in units of pixels per degree of visual angle [32]. We apply CSF to filter both the corrected benign image and adversarial example, which can be described as:

$$x' = \mathcal{F}^{-1}[H(u,v) \times \mathcal{F}[\hat{x}]], \tag{6}$$

where $\mathcal{F}[\cdot]$ and $\mathcal{F}^{-1}[\cdot]$ are the DFT and reverse DFT respectively. Note that for now, $x'$ and $x'_{adv}$ are linearly proportional to both perceived luminance and perceived contrast. The visual saliency maps are calculated as $\check{x} = x - x'$, and $\check{x}^{adv} = x^{adv} - x'^{adv}$, respectively.

*4.1.2 Structural Similarity Quantification.* After the visual saliency map is obtained, inspired by the data analysis stated in Sec. 3.2 where the structural information is shown to be able to reveal the adversarial distortion, we use the structural similarity index to quantify the perceived distortion. Specifically, it combines the luminance, contrast, and structure differences between adversarial examples and benign images and is defined as:

$$AdvDSS = [l(\check{x}, \check{x}^{adv})]^{\lambda_1} \cdot [c(\check{x}, \check{x}^{adv})]^{\lambda_2} \cdot [s(\check{x}, \check{x}^{adv})]^{\lambda_3}, \tag{7}$$

where $l$, $c$, $s$ denote the luminance, contrast, and structure comparison respectively, which can be computed from the intensity distribution as described in [11, 31]. $\lambda_1, \lambda_2, \lambda_3$ represents the weights for the three terms, where we set $\lambda_1 = 1$, $\lambda_2 = 0.48$, and $\lambda_3 = 0.48$. Please refer to Sec. 5.3.2 for the parameter adjustments.

## 4.2 Adversarial Example Quality Assessment

Here we take a step further to develop an overall quality assessment for adversarial examples. As the two crucial factors for adversarial

examples are the *invisibility* and *lethality*, we consider the perceptual quality and attack intensity as two major terms to evaluate the overall adversarial example quality. That is, a good adversarial example should have good visual quality and strong attack intensity. As we have AdvDSS as the perceptual quality index, we need to quantify the attack intensity of adversarial examples. Inspired by existing attacks [17, 36] which generates adversarial examples by maximizing the divergence from benign images, we propose to use the distribution divergence of adversarial examples and benign images in the target model as the indicator to obtain the attack intensity.

**Attack Intensity Quantification.** To quantify the distribution divergence of adversarial examples and benign images for evaluating the attack intensity of adversarial examples, we use the Jensen–Shannon (JS) divergence to measure the distribution divergence. Specifically, we first feed $x$ and $x^{adv}$ into the backbone model $f$ and obtain the logits distribution $p_x = f(x)$ and $p_{x^{adv}}$, respectively. The attack intensity of $x^{adv}$ is then computed as:

$$D_{JS} = \text{JS}(p_x \| p_{x^{adv}}), \qquad (8)$$

where the JS($\cdot$) denotes the JS divergence computation. Note that the larger JS divergence value between adversarial example and benign image denotes a stronger attack intensity since the adversarial example can easily trigger different predictions from that of benign image. An illustration is shown in Figure 5.

**Formulation of AEQA.** Combining the perceptual quality and attack intensity, the AEQA is defined as:

$$AEQA = (D_{JS})^{\alpha} (AdvDSS)^{1-\alpha}, \qquad (9)$$

where $AEQA \in [0, \infty]$ denotes the overall quality of an adversarial example. A higher AEQA value indicates a better quality of adversarial example. $\alpha$ is the weight for adjusting the impact of attack intensity and image quality. To achieve the balance between these two factors, we adopt a dynamic scheme to determine $\alpha$ as follows:

$$\alpha = \frac{1}{1 + \beta_1 (D_{JS})^{\beta_2}}, \qquad (10)$$

where $\beta_1$ and $\beta_2$ are free parameters and we empirically set $\beta_1 = 0.8$ and $\beta_2 = 0.1$. By this dynamic weight determine strategy, the AEQA pays more attention on the image perceptual quality when the value of $D_{JS}$ is large, and gives more weight on the attack intensity when the value of $D_{JS}$ is small.

## 5 EXPERIMENTS

### 5.1 Experimental Settings

**Compared Baselines.** We compare the proposed AdvDSS with extensive state-of-the-art IQA metrics on AdvDB, including FR-IQA metrics PSNR, SSIM[31], CW_SSIM[26], FSIM[39], LPIPS[40], and LPIPS-VGG[40], and NR-IQA metrics NIQE[22], BRISQUE[21], MUSIQ[15], and PI[3].

**Evaluation Criteria.** We adopted four metrics to comprehensively evaluate our approach: SROCC, KROCC, PLCC, and RMSE. SROCC and KROCC are employed to measure the prediction monotonicity of IQA methods, while PLCC and RMSE are used to measure the

prediction accuracy. Note that higher SROCC, PLCC, KROCC values and lower RMSE value denote better performance.

**Implementation Details.** Our method is implemented with Python on an NVIDIA RTX 3060Ti. In order to measure the attack intensity of adversarial examples, we use the backbone model i.e., Inception v3 network pre-trained on ImageNet, which produces the adversarial examples for AdvDB to generate the logits distribution of adversarial examples and clean images. Note that the AEQA cannot assist in the evaluation of a single adversarial example whose source model is unknown, but in the evaluation of an adversarial attack technique.

### 5.2 Main Results

**Comparisons with the Existing IQA Metrics.** We first compare our AdvDSS with the existing IQA metrics in evaluating the perceptual quality of adversarial examples. The tested SROCC, PLCC, KROCC, and RMSE results are summarized in Table 3, where we can have the following observations: 1) The NR-IQA methods including MUSIQ, PI and BRISQUE generally do not perform well for evaluating the perceptual quality of adversarial examples, as the SROCC and PLCC values of these methods for Deepfool attack are less than 0.4. There is no surprise because adversarial perturbations are always near-threshold, it is hard to directly perceive the distortions without reference images; 2) Compared to the NR-IQA methods, the FR-IQA methods perform much better since they use references. Among them, SSIM outperforms the other methods and was second only to our method, suggesting that the structural information may reveal useful perceptual attributes of adversarial perturbations; 3) The SROCC, PLCC, and KROCC values of our method are larger than all the compared methods, and for RMSE are smaller than that of the others. This denotes that, for adversarial examples, our method is better correlated with subjective evaluation scores than the existing IQA methods. The benefits come from both the detection-based scheme and structural similarity evaluation in our AdvDSS design.

We also visualize the plot distribution of objective scores versus MOS obtained on AdvDB in Figure 6. We can see that the points obtained by our proposed AdvDSS are more tightly distributed on the fitted curve than other methods, which also indicates that our method has a better consistency with the subjective evaluation scores, and is more suitable for evaluating the perceptual quality of AEs.

### 5.3 Analysis

*5.3.1 Ablation Studies.* Now we take a closer look at the design of our AdvDSS. We examine the effect of each component in AdvDSS by conducting ablation studies. Specifically, we testify AdvDSS w/o detection scheme, w/o CSF, and w/o GC. The results are reported in Table 4. As we can see, without the detection scheme, the performance drops in all metrics since the adversarial distortions are near-threshold and hard to capture. Employing the detection scheme can improve the evaluation performance, and AdvDSS achieves the best when equipped with both GC and CSF operations.

*5.3.2 Parameter Adjustments.* In our AdvDSS, the weights $\lambda_1$, $\lambda_2$, and $\lambda_3$ denote the impact of luminance, contrast, and structure in the

Table 3: Performance comparison of our AdvDSS with the state-of-the-art IQA methods for evaluating adversarial examples produced by different adversarial attacks.

| Attack | Metric | IQA methods | | | | | | | | | | | |
|---|---|---|---|---|---|---|---|---|---|---|---|---|---|
| | | PSNR | SSIM [31] | FSIM [39] | NIQE [22] | ILNIQE [38] | LPIPS [40] | LPIPS-VGG [40] | CW_SSIM [26] | MUSIQ [40] | PI [3] | BRISQUE [21] | Ours |
| PGD [18] | SROCC | 0.9268 | 0.9415 | 0.9288 | 0.9201 | 0.9415 | 0.9388 | 0.9369 | 0.9164 | 0.4988 | 0.8954 | 0.9273 | **0.9423** |
| | PLCC | 0.9675 | 0.9645 | 0.9527 | 0.9555 | 0.9646 | 0.9638 | 0.9617 | 0.9295 | 0.6488 | 0.9435 | 0.9574 | **0.9685** |
| | KROCC | 0.7775 | 0.8069 | 0.7844 | 0.7693 | 0.8070 | 0.8023 | 0.7994 | 0.7701 | 0.3389 | 0.7368 | 0.7856 | **0.8083** |
| | RMSE | 0.0922 | 0.0963 | 0.1109 | 0.1077 | 0.0963 | 0.0973 | 0.1000 | 0.1345 | 0.2776 | 0.1209 | 0.1053 | **0.0909** |
| FGSM [14] | SROCC | 0.9387 | 0.9518 | 0.9449 | 0.8869 | 0.9518 | 0.9502 | 0.9514 | 0.8972 | 0.3154 | 0.8666 | 0.8920 | **0.9583** |
| | PLCC | 0.9599 | 0.9584 | 0.9511 | 0.8638 | 0.9584 | 0.9594 | 0.9604 | 0.8817 | 0.5602 | 0.8518 | 0.8955 | **0.9685** |
| | KROCC | 0.8042 | 0.8242 | 0.8118 | 0.7194 | 0.8242 | 0.8211 | 0.8234 | 0.7395 | 0.2006 | 0.6928 | 0.7290 | **0.8373** |
| | RMSE | 0.1002 | 0.1020 | 0.1104 | 0.1801 | 0.1020 | 0.1008 | 0.0996 | 0.1687 | 0.2961 | 0.1872 | 0.1591 | **0.0890** |
| BIM [30] | SROCC | 0.9088 | 0.9397 | 0.9335 | 0.8065 | 0.9397 | 0.9370 | 0.9314 | 0.8520 | 0.5668 | 0.6977 | 0.8012 | **0.9481** |
| | PLCC | 0.9320 | 0.9510 | 0.9436 | 0.8649 | 0.9510 | 0.9486 | 0.9428 | 0.8497 | 0.5980 | 0.7539 | 0.8691 | **0.9591** |
| | KROCC | 0.7637 | 0.8042 | 0.7915 | 0.6318 | 0.8042 | 0.7978 | 0.7879 | 0.6815 | 0.4094 | 0.5376 | 0.6352 | **0.8204** |
| | RMSE | 0.1153 | 0.0984 | 0.1054 | 0.1597 | 0.0984 | 0.1007 | 0.1061 | 0.1678 | 0.2550 | 0.2090 | 0.1574 | **0.0900** |
| MIM [10] | SROCC | 0.9481 | 0.9637 | 0.9571 | 0.9052 | 0.9637 | 0.9588 | 0.9613 | 0.9216 | 0.3015 | 0.8840 | 0.9150 | **0.9674** |
| | PLCC | 0.9701 | 0.9716 | 0.9659 | 0.8877 | 0.9716 | 0.9673 | 0.9714 | 0.9047 | 0.5839 | 0.8796 | 0.9229 | **0.9764** |
| | KROCC | 0.8249 | 0.8501 | 0.8380 | 0.7479 | 0.8501 | 0.8409 | 0.8457 | 0.7777 | 0.1906 | 0.7184 | 0.7644 | **0.8592** |
| | RMSE | 0.0901 | 0.0879 | 0.0962 | 0.1710 | 0.0879 | 0.0942 | 0.0882 | 0.1582 | 0.3015 | 0.1767 | 0.1429 | **0.0801** |
| Deepfool [23] | SROCC | 0.7311 | 0.7320 | 0.7328 | 0.1355 | 0.7320 | 0.7337 | 0.7304 | 0.7221 | 0.4001 | 0.0717 | 0.0909 | **0.7338** |
| | PLCC | 0.9420 | 0.9381 | 0.9361 | 0.2582 | 0.9381 | 0.9381 | 0.9362 | 0.9049 | 0.4842 | 0.1284 | 0.3621 | **0.9356** |
| | KROCC | 0.5898 | 0.5891 | 0.5897 | 0.0991 | 0.5891 | 0.5914 | 0.5865 | 0.5820 | 0.2928 | 0.0556 | 0.0605 | **0.5923** |
| | RMSE | 0.1166 | 0.1204 | 0.1222 | 0.3357 | 0.1204 | 0.1203 | 0.1221 | 0.1479 | 0.3040 | 0.3446 | 0.3239 | **0.1227** |

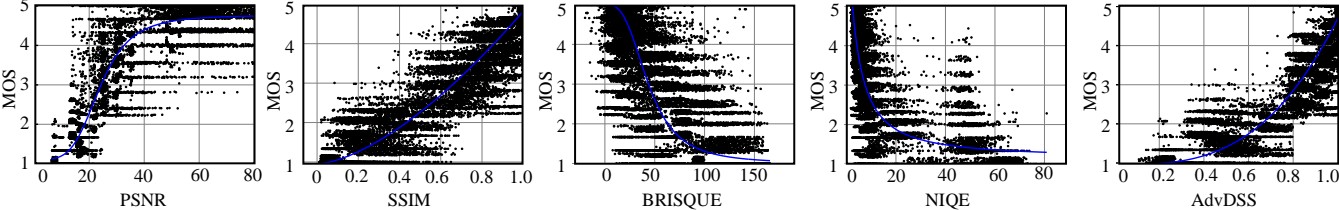

Figure 6: Scatter plots of objective IQA scores versus MOS for all the adversarial examples from our AdvDB by using PSNR, SSIM, BRISQUE, NIQE, and our AdvDSS. More visualizations are referred to Sec. C in the supplementary material.

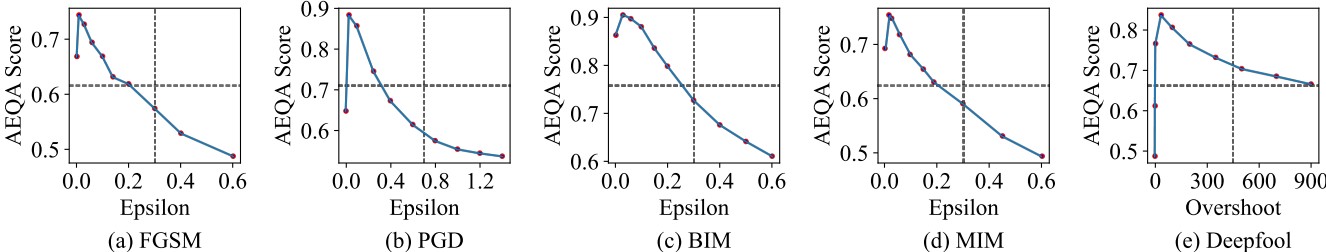

(a) FGSM    (b) PGD    (c) BIM    (d) MIM    (e) Deepfool

Figure 7: Average AEQA score curve of adversarial examples generated by different attacks with different parameter settings.

evaluation of adversarial example perceptual quality, respectively. Here we conduct experiments to discuss the efficacy of the weights in the AdvDSS. Specifically, we set $\lambda_1 = 1$, and adjust $\lambda_2$ and $\lambda_3$ to different values and show the SROCC, PLCC, KROCC and RMSE results on the AdvDB in Table 5. Note that we set $\lambda_2$ and $\lambda_3$ as the

same value to make it applicable for the computation. As we can see, the AdvDSS does not perform well at the beginning when $\lambda_2 = 0.00$ which denotes that all the three terms contribute to the AdvDSS in evaluating the perceptual quality of adversarial examples. The values of SROCC, PLCC, and KROCC increase and RMSE decreases

**Table 4: Results of ablation studies on each component of the proposed AdvDSS.**

| Method | SROCC | PLCC | KROCC | RMSE |
|---|---|---|---|---|
| AdvDSS w/o detection | 0.9057 | 0.9567 | 0.7749 | 0.1010 |
| AdvDSS w/o GC | 0.9465 | 0.9581 | 0.8136 | 0.1003 |
| AdvDSS w/o CSF | 0.9468 | 0.9579 | 0.8164 | 0.1003 |
| AdvDSS | **0.9540** | **0.9681** | **0.8313** | **0.0875** |

**Table 5: The AdvDSS performance under different values of $\lambda_2$.**

| Value of $\lambda_2$ | SROCC | PLCC | KROCC | RMSE |
|---|---|---|---|---|
| 0.00 | 0.8052 | 0.8344 | 0.6507 | 0.1906 |
| 0.20 | 0.9091 | 0.9606 | 0.7819 | 0.0957 |
| 0.40 | 0.9098 | 0.9615 | 0.7834 | 0.0945 |
| 0.44 | 0.9099 | 0.9615 | 0.7834 | 0.0945 |
| 0.48 | **0.9100** | **0.9616** | **0.7835** | **0.0945** |
| 0.60 | 0.9100 | 0.9615 | 0.7835 | 0.0948 |
| 0.80 | 0.9100 | 0.9610 | 0.7834 | 0.0954 |
| 1.00 | 0.9099 | 0.9603 | 0.7833 | 0.0964 |

**Table 6: Average AEQA scores of different attacks. The attack settings are the same as we use $L_2$-norm and the $\epsilon$ is set to be 0.03 for fair comparison.**

| Attack technique | AEQA | AdvDSS | $D_{JS}$ |
|---|---|---|---|
| BIM [30] | 0.8747 | 0.9780 | 0.8096 |
| PGD [18] | 0.8570 | 0.9761 | 0.7673 |
| FGSM [14] | 0.7442 | 0.9386 | 0.6422 |
| MIM [10] | 0.7346 | 0.9510 | 0.5892 |

as the value of $\lambda_2$ increases. The peak value appears when $\lambda_2 = 0.48$. We adopt this setting throughout the whole experiments.

## 5.4 Applications and Analysis

*5.4.1 Comparison of Adversarial Attack Quality using AEQA.* We compare the quality of AEs produced by different adversarial attacks using the proposed AEQA. To make a fair comparison, we select AEs generated of different attack techniques with the $L_2$-norm as 0.03. The AEQA scores and their components are summarized in Table 6. In general, the PGD and BIM achieve comparable performance as BIM surpasses PGD by 0.02 in terms of AEQA. In contrast, FGSM has the lowest AEQA due to its weak attack intensity and poor perceptual quality.

*5.4.2 Choosing the Best Settings for Each Attack using AEQA.* It is always hard to determine the best parameter settings, *i.e.*, the step $\epsilon$ in adversarial attacks, as a larger value of $\epsilon$ leads to a stronger attack intensity but can also induce severe distortions in adversarial examples. Here we try to explore the best setting in each attack

**Table 7: The attack performance of AEs generated by directly optimizing AEQA on different datasets.**

| Dataset | CIFAR-10 | CIFAR-100 | Tiny-ImageNet |
|---|---|---|---|
| Attack success rate | 0.9596 | 0.9437 | 0.9243 |

with AEQA score. Specifically, we compute the average AEQA score of adversarial examples at different parameter settings. The results are shown in Figure 7. As we can see, the AEQA score of each attack increases as the $\epsilon$ or *overshoot* increases and decreases before it reaches the peak value. PGD and BIM have higher peak value than the other three attacks because they can generate AEs with better quality as validated in Table 6. For practical use, we can adopt the best parameter setting for generating adversarial examples with best quality.

*5.4.3 Generating Adversarial Examples using AEQA.* We also conduct an interesting experiment in which we try to generate AEs by directly optimizing the AEQA. Specifically, we set the AEQA as the objective and use L-BFGS algorithm to directly generate an adversarial example from a clean image with high AEQA score. We use the Inc V3 pre-trained on ImageNet as the target model and also the backbone to compute JS-Distance. We set the iteration step as 20 during optimization. The attacks are performed on different datasets and the results are reported in Table 7. It can be observed that the AEs exhibit a high attack success rate on all the datasets. Due to the page limitation, more details and qualitative results are referred to our supplementary material.

## 6 CONCLUSION AND DISCUSSION

In this paper, we have constructed the first and largest adversarial example database, namely AdvDB, for assessing the perceptual quality of adversarial examples. Based on AdvDB, we have proposed a detection-based quality assessment approach, AdvDSS, which employs a visual salience detection scheme to first obtain the salience map from the HVS perspective and then quantify the distortions by using structural similarity between adversarial examples and benign images. Also, we take a step further to propose AEQA to conduct an overall quality assessment of adversarial examples by integrating the perceptual quality and attack intensity. The experiments demonstrate that our method can achieve superior performance than the existing IQA methods.

**Discussion.** In this work, the AdvDSS is performed in a traditional HVS manner and did not employ deep models for two reasons: 1) The adversarial examples have the property of *transferability* that can potentially disturb other deep models; 2) The proposed method is a simple yet powerful benchmark method that can be easy to apply, which we believe can be set as a baseline for future research.

**Limitations.** The proposed database only contains gradient-based attacks since they are the most effective attack strategies for now. The MOS annotations can also be manually decomposed into diverse attributes such as contrast, color, and saturation for deep analysis.

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
