# OpenReview forum: "Adversarial Example Quality Assessment: A Large-scale Dataset and Strong Baseline"
_acmmm.org/ACMMM/2024/Conference — MM2024 Poster_

### Official Review · Reviewer_EuTW · 2024-04-29

**Rating:** 3
**Confidence:** 3

**Summary:**

This paper introduces a dataset named AdvDB, which comprises 18,346 adversarial examples generated using five different attack methods, complete with annotations. Building on this dataset, the authors develop AdvDSS, a system designed to capture the characteristics of adversarial perturbations. Additionally, they propose integrating AdvDSS with the Jensen-Shannon divergence to create AEQA, a new quality assessment metric designed specifically for evaluating adversarial examples.

**Strengths:**

(1) This paper focuses on the image quality assessment of adversarial examples, a topic that has not yet been extensively explored within the community.

(2) The proposed AdvDSS system demonstrates performance that is slightly superior to existing full-reference image quality assessment (FR-IQA) metrics, such as LPIPS.

(3) In my opinion, the most significant contribution of this work is the annotation of the MOS for adversarial examples through subjective experiments. It experimentally demonstrates that existing FR-IQA metrics correlate well with the MOS. Therefore, it is reasonable for researchers in the attack community to use existing FR-IQA metrics.

**Limitations:**

Some main questions about this work:

Q1: About the dataset. As the paper illustrated, there are totally #total=1500(samples)*5(attack methods)*5(parameters choices) adversarially perturbed images in therotical. However, only 18,346 images successfully attacked the backbone, so the attack success rate is only about 18,346/#total < 50%. Even without considering cases where $\epsilon < 0.01$, the attack rate is about 60%. This seems unusual, as all the attack methods used in this paper are generally known to achieve over 90% attack sucess rate when $\epsilon \geq 0.03$.

Q2: About the meaning of AdvDSS. As the paper shows in Fig.3 and Tab.3, exsting FR-IQA metric, such as LPIPS and LINIQA, align well the human perceptual system when assessing the quality of adversarial examples. The proposed AdvDSS only marginally improves the SROCC (from Tab. 3, it increases SROCC from LINIQE-0.9415 to 0.9423), KROCC, PLCC and the RMSE. I believe the improvement is minimal. In other words, existing models could suffice for evaluating adversarial samples.

Q3: The motivation behind AEQA is unclear. According to the authors, AEQA is not applicable if the source model is unknown, suggesting that AdvDSS might be more broadly applicable. It raises the question: In what scenarios is AEQA more useful than AdvDSS? This needs to be addressed to understand the distinct advantages of AEQA over AdvDSS.

Q4: The results shown in Fig.7 are puzzling. The AEQA scores for clean images (epsilon=0) are lower than those for images with small adversarial perturbations (epsilon=0.03), suggesting that the quality of clean images is perceived to be worse than that of slightly perturbed adversarial examples. This seems counterintuitive and potentially incorrect. Even if I misunderstood and the Fig.7 does not include epsilon=0, the conclusion that the AEQA score increases and then decreases with larger epsilon values still seems strange to me.

Other minor questions that do not afftect my rating:

M1: In calculating RMSE in Tab.3, how do the authors map the scores from existing FR-IQA metrics to the MOS range? For instance, SSIM is scaled between [0,1], but the MOS collected in this work ranges from [1,5]. I think the RMSE is related with the mapping.

M2: Some typos in the paper. (1) Tab.2, all methods are incorrectly marked the same in the column $L_p$ norm. (2) Line 344(right), I think there are only 25 attack strategies, not 75. (3) Tab.6, in the caption, $L_2 -> L_\infty$. (4) Fig.3(b), the score range should be left-closed and right-open, e.g. $BIM[1,2), BIM[2,3)$.

**Suitability:**

2

---

### Official Review · Reviewer_2DUD · 2024-05-23

**Rating:** 4
**Confidence:** 2

**Summary:**

The paper proposes a novel benchmark adversarial example quality assessment metric for comprehensively evaluating the quality of adversarial examples.

**Strengths:**

The paper demonstrates the SOTA performance of the proposed adversarial example quality assessment metric through large-scale experiments. The paper also generates better adversarial examples based on this metric.

**Limitations:**

The experiment only includes gradient based attacks, and these attacks are relatively old. I hope the experiment can supplement some other attacks, such as optimization based or frequency domain based, etc. I am confused that CIFAR-10/100 was also used for dataset construction, and the number of pixels in these datasets is very small. Participants may have difficulty distinguishing the adversarial perturbations, especially when the metrics have five levels. This paper focuses on the issue of single media and does not align well with the theme of multimedia.

**Suitability:**

2

---

### Official Review · Reviewer_kUFM · 2024-05-23

**Rating:** 4
**Confidence:** 3

**Summary:**

This paper presents a large-scale adversarial dataset, AdvDB, to serve as a foundation for image quality assessment for adversarial examples (AEs), a detection-based structural similarity index, AdvDSS for AR perceptual quality assessment, and a combinational index, AEQA, which measures the quality of AEs from the perspectives of perception quality and attack intensity. The authors conduct experiments with 5 attack methods, 11 baseline IQA metrics, and 4 evaluation criteria. The experiment results show that AdvDSS outperforms the baseline metrics among all adversarial attacks and evaluation criteria.

**Strengths:**

+ Timely and interesting problem.
+ New dataset and perceptual quality assessment index for AEs, which may benefit the community.
+ Large-scaled experiments with various baseline metrics, evaluation criteria and attack methods.

**Limitations:**

- Novelty requires further clarification.
- Lack of justification for some technical details.
- The experiments for AEQA seem incomplete.

### Detailed comments:

1. Some related works have studied the adversarial examples with visual quality[1-3], although few works systematically investigate the evaluation of AEs from a perceptual quality perspective. The authors propose AdvDss, a new index to measure the corresponding perceptual quality; however, to me, it is more like a combination of existing techniques. The authors may further highlight the novelty of such an index design.

2. The authors select 5 methods for adversarial example generation; nevertheless, there are some concerns regarding the selection criteria. In particular, it is not clear why selecting these 5 methods for generation; are they representative and/or commonly used? Besides, some of these methods do not seem to be state-of-the-art AE generation approaches; it is vital to justify the rationale behind such selection. Also, the authors mentioned that they only took the gradient-based attacks for the experiment; however, I would recommend including optimization and learning-based methods to deliver a more generalized conclusion.

3. Some formulas need further explanation. The authors define many nuanced parameters to facilitate the design of the similarity index (i.e., equations 7, 9, 10). There is a lack of details about the design scheme of these parameters, which may confuse readers who are not familiar with this field.

4. The authors propose two evaluation indexes, AdvDSS and AEQA, to facilitate the quality measurement of AEs from a perceptual quality aspect and an integrated aspect, respectively. Regardless of the experiments for AdvDSS, the paper does not provide a comprehensive evaluation of AEQA. The authors may discuss more about the effectiveness of AEQA among different attack methods (like Table 6) and justify the trade-off between the perceptual quality assessment and the attack intensity assessment, namely, AdvDSS and Djs.

5. The manuscript is overall written in a motivating and systematic manner.


[1] "Improving the invisibility of adversarial examples with perceptually adaptive perturbation." Information Sciences 635 (2023).

[2] "Chaotic Variational Auto encoder-based Adversarial Machine Learning." (2023).

[3] "Improving perceptual quality of adversarial images using perceptual distance minimization and normalized variance weighting." In The AAAI-22 Workshop on Adversarial Machine Learning and Beyond. (2021)

**Suitability:**

3

---

### Meta-Review · Area_Chair_vCrA · 2024-07-01

**Recommendation:** Accept (Poster)
**Confidence:** 5

**Metareview:**

A rebuttal letter is submitted, and the reviewers acknowledged the clarification and update for the paper. All reviewers merge to the rating of borderline accept by considering the work novelty and thorough experiment design and evaluation. However, more experimental details will be added and further clarification of novelty will be updated in the camera-ready version. A decision of accept is suggested.